# A Concept for Bio-Agentic Visual Communication: Bridging Swarm Intelligence with Biological Analogues

**DOI:** 10.3390/biomimetics10090605

**Published:** 2025-09-09

**Authors:** Bryan Starbuck, Hanlong Li, Bryan Cochran, Marc Weissburg, Bert Bras

**Affiliations:** 1George W. Woodruff School of Mechanical Engineering, Georgia Institute of Technology, Atlanta, GA 30332, USA; bryan.cochran@gatech.edu (B.C.); bert.bras@me.gatech.edu (B.B.); 2School of Biological Sciences, Georgia Institute of Technology, Atlanta, GA 30332, USA; marc.weissburg@biology.gatech.edu

**Keywords:** biologically inspired communication, generative AI, large language model, retrieval-augmented generation, unmanned aerial vehicle swarms, agentic AI, multi-agent system

## Abstract

Biological swarms communicate through decentralized, adaptive behaviors shaped by local interactions, selective attention, and symbolic signaling. These principles of animal communication enable robust coordination without centralized control or persistent connectivity. This work presents a proof of concept that identifies, evaluates, and translates biological communication strategies into a generative visual language for unmanned aerial vehicle (UAV) swarm agents operating in radio-frequency (RF)-denied environments. Drawing from natural exemplars such as bee waggle dancing, white-tailed deer flagging, and peacock feather displays, we construct a configuration space that encodes visual messages through trajectories and LED patterns. A large language model (LLM), preconditioned using retrieval-augmented generation (RAG), serves as a generative translation layer that interprets perception data and produces symbolic UAV responses. Five test cases evaluate the system’s ability to preserve and adapt signal meaning through within-modality fidelity (maintaining symbolic structure in the same modality) and cross-modal translation (transferring meaning across motion and light). Covariance and eigenvalue-decomposition analysis demonstrate that this bio-agentic approach supports clear, expressive, and decentralized communication, with motion-based signaling achieving near-perfect clarity and expressiveness (0.992, 1.000), while LED-only and multi-signal cases showed partial success, maintaining high expressiveness (~1.000) but with much lower clarity (≤0.298).

## 1. Introduction

Biological swarms exhibit complex, adaptive behavior through local interactions rather than centralized control. In systems like bee colonies, fish schools, and ant trails, individuals make autonomous decisions using limited cues and internal thresholds. Behavior is shaped by salience, selective attention, and feedback loops that balance reinforcement and inhibition. For example, not every bee follows its neighbors’ waggle dance; each evaluates the signal in context to avoid overcommitment. Organisms do not respond equally to all neighbors but prioritize certain cues. Some species use direct displays, such as gestures or movement, while others modify their environment with chemical trails. These principles of animal communication [1] support scalable coordination in dynamic environments.

The objective of this proof of concept is to demonstrate how these biological principles can be translated into swarm-based visual communication for unmanned aerial vehicles (UAVs) using Generative AI (GenAI). The complexity and context dependence of bio-strategies make them difficult to encode. However, GenAI enables learning of symbolic and interpretable patterns from bio-inspired exemplars, where interpretability ensures that a signal has clarity (ease of recognition and understanding), and learning new symbols ensures that a language may be expressive (adaptive and rich) enough to communicate nuanced details effectively. These models generate new behaviors in real time, adapting to partial local input, a capability not achievable with conventional AI or genetic algorithms. With this approach, UAV agents evolve a baseline visual language that combines multiple bio-strategies into an interpretable, multi-cue signal.

This translation is timely as low-cost adversarial UAVs are increasingly used in conditions where they are hard to detect or counter, such as radio frequency (RF)-denied environments [2]. Interceptor swarms offer defense [3], yet lack the ability to communicate when wireless communication is denied. This work focuses on multi-rotor UAVs, which provide precise omnidirectional control over translation and rotation so a broad range of bio-strategies could be communicated visually through motion signals.

The problem that this proof of concept addresses is that current UAV swarms remain dependent on RF links, which are vulnerable to jamming and spoofing [4], and centralized control systems, which scale poorly and are fragile in contested environments [5]. Biologically inspired visual signaling supports RF-free formation control [6] and allows UAV agents to communicate using LED patterns, in addition to motion. Optical channels can offer high data rates, resist interference [7], and enable covert, spectrum-free signaling [8]. However, most existing visual systems use static gestures or codes, limiting adaptability in dynamic scenarios [9]. This rigidity contrasts with natural signaling, where communication strategies are inherently adaptive, context-sensitive, and capable of evolving in response to changing conditions.

Thus, biological systems provide a richer model. For example, bees use waggle dancing to encode location, fireflies synchronize by flashing, and wolves coordinate to encircle their prey [10]. Robotic studies show that even simple LED and motion rules can produce emergent coordination [11]. Advances in GenAI, particularly large language models (LLMs) with retrieval-augmented generation (RAG), now make it easier to learn emergent languages. These models can develop symbolic communication [12], bypassing rigid protocols. They can also shift contextual modalities to maintain swarm coherence [13]. Recent work on autonomous agent swarms further highlights how decentralized, adaptive coordination enables scalable and resilient group behavior, aligning with these biologically inspired approaches [14].

While advances in AI and decentralized coordination address the problem in part, we introduce two novel contributions not present in existing work. First, no prior method preconditions a GenAI model, specifically an LLM with retrieval mechanisms, on biologically inspired visual signaling. Second, no known system uses a maneuver and LED-based token structure to represent and execute swarm behavior through LLM outputs. In this approach, a GenAI translation layer is preconditioned with a curated set of biological signaling strategies. The model maps perceptual inputs into symbolic responses that are expressed through changes in position, orientation, and LED state. A custom tokenization scheme encodes these visual behaviors into structured messages that convey intent, role, or alert information. Communication arises not from fixed rules but from learned, contextual mappings, allowing UAVs to coordinate using only local perception. Because each UAV operates as an autonomous agent in a multi-agent system, this proof of concept exemplifies agentic AI, where each agent maintains role-awareness, interprets peer signals, and generates responses through reasoning. Coordination thus emerges from agent interaction protocols in the form of biological signaling, and system performance is evaluated by measuring the effectiveness of inter-agent communication in producing coherent and adaptive swarm behaviors.

Five biologically inspired test cases validate this system, including scenarios based on bee waggle dances, deer tail-flagging, and peacock status displays. Results demonstrate that the system reliably preserves symbolic intent through within-modality fidelity and integrates multiple cues in more complex situations via cross-modal translation. This indicates not only a capacity for clarity and consistency, but also for expressive signal adaptation when multiple forms of information are present. By combining biological insight with generative reasoning, this work establishes a foundation for decentralized, interpretable, and robust swarm communication under contested conditions. Figure 1 illustrates the contrast between RF-disrupted communication and bio-agentic visual interception.

The remainder of this article is structured as follows:▪Section 2 (Literature Review): Summarizes key biological analogues and their computational counterparts, and provides a deep synthesis of swarm intelligence principles and biological communication models.▪Section 3 (Methods): Describes the GenAI layer, including bio-inspired strategy identification, selection, model architecture, mathematical formulation, and experimental setup.▪Section 4 (Results): Presents the results and discussion of the test cases to evaluate the visual communication system for UAV swarm agents.

## 2. Literature Review

To contextualize our approach, Table 1 presents a structured review of biological communication strategies and their computational counterparts, organized by UAV communication functionality. For each function, we identify conventional limitations in RF-based or centralized systems, highlight emerging swarm solutions, and draw explicit parallels to biological analogues. This framework positions our method within a broader convergence of swarm intelligence principles, local decision-making, salience, symbolic signaling, and adaptive coordination, showing how nature-inspired strategies can inform computational techniques that enable decentralized, expressive, and clear UAV swarm communication.

Conventional systems rely on RF commands to convey spatial intent, but these are fragile in contested environments and lack adaptability. Recent GenAI methods allow UAVs to encode intent directly into motion [15], translating linguistic goals into directionally coordinated trajectories [16] and coordinated flight paths for group cohesion [17]. These techniques parallel biological behaviors such as bee waggle dances, which encode direction and distance to a specific goal or location [18], ant trails, which guide movement via pheromone concentration [19], and archerfish line-of-sight targeting [20], where motion replaces continuous signaling with shared, embodied cues. In computational terms, the bee waggle becomes trajectory tokens in GenAI outputs because both can encode direction and distance as symbolic motion vectors. Ant pheromone trails map to reinforcement-style signaling, where repeated traversal strengthens the chemical pathway through increased pheromone concentration, making it progressively easier for subsequent agents to detect and follow. Archerfish targeting parallels trajectory prediction models, as both rely on precise line-of-sight estimation to project future states and guide coordinated action. Each of these aligns with swarm intelligence principles of decentralized control (decisions made without central command), interactions made via the environment (neighbors follow trails or trajectories as others leave scent or markers), and salience (signals prioritized when direction or target information is critical). Taken together, these analogues and their computational counterparts show how directional encoding is achieved through local, interpretable motion cues that support scalable coordination.

RF-based threat alerts and centralized awareness degrade under jamming or partial observability, but multi-agent reinforcement learning (MARL) agents evolve symbolic alerts under partial observability [21], maintain robust learned protocols amid environmental variability [22], and adapt communication in adversarial environments [23], shifting communication toward local, adaptive behaviors. This mirrors biological warning cues, such as tail-flagging in deer, which signals threats to group members [24], echolocation in bats, which doubles as an obstacle alert mechanism [25], and pronking in gazelles, which increases visibility and signals danger [26], where fast signals convey danger without centralized control. The deer tail flag corresponds to binary symbolic LED alerts in UAVs, with both acting as on–off signals under threatening conditions. Bat echolocation resembles sonar-like obstacle sensing integrated into MARL policies, where returning signals update and reward local awareness in real time. Gazelle pronking parallels high-visibility trajectory shifts in UAV swarms, as both serve as exaggerated movement patterns that communicate urgency to the group while simultaneously deterring adversaries. These capture swarm intelligence principles of selective attention (agents prioritize high-salience alerts), robustness (alerts propagate despite degraded channels), and emergent coordination (groups align responses without central command). Thus, biological warning signals and computational MARL alerts converge on the principle that threat information must propagate rapidly and robustly using lightweight symbolic cues.

Role assignment in traditional systems depends on centralized coordination, limiting flexibility in dynamic or RF-constrained settings. New approaches use MARL for local role adoption [27], LLMs for dynamic role switching [28], and decentralized messaging for autonomous tasking [29]. Biological systems show similar distributed strategies, from lion ambush coordination through role-based movement [30] to peacock feather displays as visual status cues [31] and ant task switching based on local demands [32], where visual or spatial cues support decentralized role management. Lion ambush coordination maps to MARL agents negotiating roles through local reinforcement, where distributed signals determine who advances or holds back. Peacock feather displays parallel LLM-driven symbolic role signaling via LEDs, with both using conspicuous, persistent cues to communicate social status or task assignment. Ant task switching reflects dynamic role reassignment algorithms, with local thresholds governing flexible task allocation that mirrors decentralized computational role adoption. These examples illustrate swarm intelligence principles of decentralization (roles emerge locally), context-dependence (agents switch roles as needed), and feedback loops (role persistence or change based on environmental reinforcement). Together, these strategies demonstrate how distributed role differentiation in biology translates into adaptive, scalable role assignment in UAV swarms.

Collision avoidance in conventional swarms is centralized and brittle. In contrast, recent work leverages shared trajectories for real-time avoidance [33], AI for predictive spacing and coordination [34], and MARL for formation integrity [35]. These strategies parallel animal flocking and schooling behaviors, such as starling alignment to prevent collisions [36], goose spacing in energy-efficient formations [37], and yellowtail fish spacing through local sensing [38], where feedback ensures cohesion. Starling alignment is mirrored in predictive trajectory alignment algorithms, where each agent adjusts based on neighbors to prevent collisions. Goose V-formations correspond to optimization models that balance energy efficiency with spacing, with both exploiting the aerodynamic or algorithmic benefits of structured formations. Yellowtail fish sensing parallels local sensing modules in MARL spacing policies, where simple perception to response rules maintain safe yet cohesive inter-agent distances. These directly express swarm intelligence principles of emergent coordination (flocking rules aggregate into global order), scalability (rules hold as numbers increase), and redundancy (local sensing ensures robustness to failures). Thus, biological flocking and computational collision avoidance share a foundation of local feedback loops that scale to resilient group cohesion.

Fragmentation following collisions or interference often breaks traditional swarm cohesion. Bio-inspired approaches use symbolic UAV motions for regrouping [39], digital trail emulation for passive reformation [40], and MARL for restoring shared intent [41]. Nature offers analogous behaviors, such as sardine bait ball reformation after predator disruption [42], bird-inspired optical strategies for regaining formation [43], and starling mid-flight adjustments [44], all relying on local visual cues rather than communication links. Sardine regrouping aligns with MARL reformation protocols, where swarm agents converge after dispersal using shared intent cues. Bird optical signals parallel LED-based rejoin mechanisms in UAVs, with both employing conspicuous visual flashes to restore group cohesion. Starling mid-flight adjustments correspond to trajectory re-synchronization via AI-generated corrective motion, highlighting how local, small-scale adaptations sustain swarm integrity after disruption. These reflect swarm intelligence principles of robustness (recovering from disruption), salience (visual cues used to prioritize rejoining), and decentralized adaptation (each agent re-aligns without central command). This synthesis highlights how both biology and computation use local visual cues to ensure swarm integrity under fragmentation.

Conventional mission updates depend on RF signals, which fail under bandwidth constraints. New methods replace these with formation and role shifts to reflect mission phases [45], trajectory morphing to encode outcomes [46], and visual cues to signal progress without central control [47]. This mirrors structured biological signaling, such as elephant trunk gestures [48], bowerbird display escalation to show readiness [49], and tree frog call changes [50], where behavioral phases are visually marked. Elephant trunk gestures correspond to UAV trajectory morphing as both act as symbolic signals that mark distinct phases of collective activity. Bowerbird display escalation parallels the gradual intensification of UAV LED patterns, where increasing salience communicates readiness or role change. Similarly, frog call variations map to phase-coded motion or lighting tokens in UAV swarms, with both functioning to segment behavioral phases and synchronize group transitions. These embody swarm intelligence principles of symbolic signaling (clear encoding of phases), adaptability (signals evolve with mission state), and scalability (individual cues scale up to group-level mission coherence). Thus, structured biological signaling and computational phase encoding both enable decentralized mission progression without RF.

Single-point RF systems are vulnerable to jamming or node failure. Decentralized swarm protocols now use distributed alerts for overlapping coverage [51], shared observations for swarm awareness [52], and trajectory cues for early warnings [53], achieving redundancy through overlapping cues. These designs reflect biological fault tolerance, including ant pheromone trail reinforcement [54], intermittent firefly flashing [55], and multi-sentinel bird coordination [56]. In computational systems, MARL-based distributed alert mechanisms reflect the reinforcement of ant pheromone trails, where overlapping signals preserve robustness despite individual failures. Asynchronous signaling protocols resemble firefly flashing, ensuring that swarm members can resynchronize even if some cues are missed. Likewise, multi-agent redundancy strategies mirror sentinel bird rotations, where responsibility for vigilance is distributed, minimizing vulnerability and maintaining continuous group awareness. These highlight swarm intelligence principles of redundancy (overlapping signaling ensures resilience), local interactions (agents update state from immediate neighbors), and robustness (system persists despite node failures). This synthesis illustrates that both biological and computational systems achieve resilience through overlapping, decentralized cues.

Conventional swarms struggle to adapt in real time. Anticipatory strategies now allow UAVs to reallocate tasks and reroute after failures [57], anticipate adversarial behavior using MARL [58], and align trajectories through prediction [59]. These methods resemble natural anticipation behaviors, such as wolf encirclement based on predicted escape routes [60], orca synchronization to block prey [61], and baboon alignment through neighbor projection [62], where coordination emerges through prediction rather than fixed commands. MARL predictive encirclement mirrors wolf pack hunting, where agents anticipate prey escape routes and adjust positions collectively. Orca synchronization maps to UAV trajectory convergence, as both emphasize coordinated timing to block an adversary’s movement. Baboon neighbor projection aligns with predictive alignment models, with both relying on projecting local neighbor motion into future states to maintain group cohesion and preempt fragmentation. These link directly to swarm intelligence principles of emergent coordination (group strategy emerges from local predictions), adaptability (agents anticipate and adjust in real time), and decentralized decision-making (predictions made without central control). Thus, anticipatory strategies in both biology and UAVs reinforce the value of prediction-based local responses for adaptive coordination.

This review highlights how biological analogues and their computational counterparts converge on core swarm intelligence principles: decentralized control, local interactions, selective attention to salient cues, and the balance of reinforcement and inhibition through feedback loops. Directional cues from behaviors such as bee waggles or ant trail following illustrate how local signals guide the trajectories of subsequent agents, even when the exact motion path is not explicitly encoded. Threat alerts (deer flagging, bat echolocation) demonstrate rapid propagation of high-salience signals. Role assignment (lion ambush, peacock displays) highlights adaptive differentiation without centralized command. Flocking and reformation (starlings, sardines) emphasize emergent coordination and robustness. Structured signaling (elephants, frogs) ensures interpretable mission phases. Redundancy (fireflies, sentinel birds) secures resilience. Anticipation (wolves, orcas) underscores prediction-driven coordination. Together, these analogues establish a conceptual framework where biologically inspired communication maps systematically onto decentralized UAV control mechanisms through bio-agentic swarm communication.

## 3. Methods

This methods section presents an in-depth overview of the proposed bio-agentic visual communication approach for UAV swarms operating in RF-denied environments. It begins by first identifying biologically inspired signaling strategies through a morphological analysis based on Encoding Precision and Role Cue Clarity, Maneuver and Display Map-ability, Symbolic Simplicity, and Interoperability for Group Response (Section 3.1). A reference architecture is then introduced (Section 3.2), where each UAV uses a perception module, a GenAI layer, and a bio-strategy database to interpret peer behaviors and generate visual responses. These strategies are encoded into an extended configuration space that includes position, orientation, and LED states over time, forming a structured input-output format for learning and generation (Section 3.3). Section 3.4 details a set of test cases and evaluates response fidelity to measure how well UAVs preserve or translate observed signals. Lastly, Section 3.5 describes the algorithm that guides the swarm’s visual reasoning, combining structured prompts, biological templates, and language model outputs to support symbolic, multimodal communication.

### 3.1. Selected Principles of Communication

To bridge the design gap between biological analogues and engineered swarm communication, we began with a functional decomposition [63] of communication types in the literature review. This helped determine and isolate the following key criteria, aligned with swarm communication principles, for evaluating biological strategies on their translatability to UAV swarm communication. Encoding precision and role cue clarity relate to salience and selective attention, because accurate directional or role cues ensure that agents prioritize the most relevant signals while filtering noise. Maneuver and display map-ability reflects local interactions and embodied signaling, given that clear mapping of gestures or trajectories into visual/motion space enables neighbors to directly interpret and respond without centralized instructions. Symbolic simplicity aligns with symbolic signaling and inhibition of overcommitment, as simpler signals reduce ambiguity and prevent agents from overreacting to weak or conflicting cues. Finally, group response interoperability captures feedback loops and decentralized coordination, given that the ability of a signal to propagate and trigger collective action depends on its compatibility with distributed decision-making.

We then selected three foundational contexts for communication within the threat interception use case: Target Location, Obstacle Alerts, and Role or Status, as a proof of concept, noting that this analysis is not meant to be exhaustive at this stage of development. These contexts represent high-priority, generalizable swarm behaviors that can be clearly conveyed through movement or LED-based cues. We then constructed a morphological chart to map biological analogues to each function, following standard practice for designing bio-inspired visual communication systems [64].

We evaluated bio-strategy translatability using a qualitative checklist rubric (✓ = strong alignment, ~ = partial alignment, × = weak alignment) in Table 2. In this context, strong alignment (✓) means the biological signal fully satisfies the key criteria. Partial alignment (~) indicates that the criteria is only partially satisfied, often because the signal works in nature but loses fidelity or clarity when mapped into UAV contexts. Weak alignment (×) indicates a lack of support in UAV setting, typically from modality mismatch or poor scalability. This rubric allows systematic comparison of nine biological strategies across the three contexts, highlighting those most likely to be transferable to UAV swarms.

The Honeybee waggle dance is the most effective biological strategy for encoding spatial objectives in UAV swarms due to its precision, symbolic structure, and visual adaptability. Forager bees perform a rhythmic dance encoding direction via angle and distance via duration, allowing others to locate a food source without direct line-of-sight [65]. Research has shown that experienced bees adjust their encoding based on prior knowledge, overriding misleading optic cues. This makes the waggle dance a case of strong alignment ✓ in encoding precision and role cue clarity, since the signal directly and reliably encodes directional vectors. Its oscillatory path also translates into UAV maneuvers, giving it strong alignment ✓ in maneuver and display map-ability. Because the dance is rhythmic, repeatable, and interpretable as a compact code, it also achieves strong alignment ✓ in symbolic simplicity. Finally, because it propagates easily across swarm members once observed, it provides strong alignment ✓ in group response interoperability, making it one of the most transferable strategies overall.

In contrast, ant pheromone trails guide movement through chemical gradients [66] but encode directionality only indirectly, since ants determine orientation by sampling concentration gradients along the trail rather than receiving an explicit signal, resulting in weak alignment × in encoding precision. While trails can persist and reinforce collective movement (where the trail itself becomes reinforced by repeated use, offering partial alignment ~ in interoperability), they are difficult to translate into UAV motion or visual cues, producing weak alignment × in maneuver-display mapping and weak alignment × in symbolic simplicity. This is due to the fact that, in the context of signaling, UAVs cannot modify the environment in a way that provides information, nor can they reliably decode or detect the environmental modifications that would provide information. This is the same reason why flocking and swarming that is based on hydrodynamic or fluid dynamic signals show poor alignment, as UAVs similarly lack the ability to decode these signals.

Archerfish use precise line-of-sight aiming, adjusting for physical conditions like refraction and wind [67], which achieves strong alignment ✓ in encoding precision, but the requirement of a visible target makes it less adaptable in swarm contexts. This yields only partial alignment ~ in maneuver-display mapping and weak alignment × in both symbolic simplicity and interoperability. The waggle dance offers symbolic, repeatable motion that can be mapped to UAV trajectories and LED pulses, making it uniquely suited for spatial coordination in visual communication environments.

White-Tailed Deer tail-flagging provides a simple, high-visibility alarm signal ideal for UAV obstacle alerts. When threatened, a deer lifts its tail to expose a bright white patch, signaling others to flee and potentially deterring pursuit [68]. This binary but effective signal achieves partial alignment ~ in encoding precision (since it does not encode detail but reliably indicates danger), and strong alignment ✓ in symbolic simplicity, as it is both conspicuous and unambiguous. Its direct translation into LED flashes also gives strong alignment ✓ in maneuver-display mapping, and its ability to rapidly propagate through a group offers strong alignment ✓ in interoperability.

In contrast, bat echolocation enables detailed individual navigation through ultrasonic calls and echo processing [69]. While this provides strong alignment ✓ in encoding precision, it is weak alignment × in maneuver-display mapping and weak alignment × in symbolic simplicity, since it cannot be readily visualized. It also provides weak alignment × in interoperability, because it is more suited for individual navigation than swarm-level communication.

Gazelle pronking involves high leaps that signal fitness and alert others [70], which demonstrates only partial alignment ~ in encoding precision, since the display combines multiple highly specific motion elements (height, posture, timing) and is not easily distinguishable from other potential movements. It also requires energetically costly vertical maneuvers, giving only partial alignment ~ in maneuver-display mapping. Because of this, it scores weak alignment × in symbolic simplicity and interoperability, making it less transferable to UAV signaling. Tail-flagging balances visibility, simplicity, and interpretability, offering a direct visual warning that is easy to implement and propagate through a swarm.

The peacock feather display is the best fit for conveying role or status in UAV swarms due to its persistent, symbolic, and visually expressive design. Males display iridescent eye-spots while vibrating their trains, creating a shimmering effect that signals fitness and dominance [71]. This behavior provides partial alignment ~ in encoding precision, since it conveys relative quality rather than explicit vectors, but it achieves strong alignment ✓ in maneuver-display mapping when adapted as rhythmic LED color patterns. Its distinctiveness also supports strong alignment ✓ in symbolic simplicity, and because the display is easily interpretable and persists across interactions, it achieves strong alignment ✓ in interoperability. In UAVs, this can be mapped to unique LED colors or pulse patterns that persist across maneuvers, making roles visually distinguishable.

Lions coordinate ambushes using learned positional roles like wings and centers [72], providing strong alignment ✓ in encoding clarity but only partial alignment ~ in maneuver-display mapping, since roles are inferred from positioning rather than explicit signals. Lions show weak alignment × in symbolic simplicity and interoperability, as their strategies lack persistent, shareable markers.

Ants allocate tasks through local interactions and encounter rates [73], giving strong alignment ✓ in encoding precision and interoperability, but they show weak alignment × in symbolic simplicity and maneuver-display mapping, since the cues they rely on do not provide instantaneous direction or distance information. Instead, they depend on stepwise repeated sampling, where early decisions may be incorrect before convergence on the correct path gradually occurs. The peacock model uniquely supports stable, role-specific signaling through color and rhythm, aligning well with swarm-level visual communication needs.

This evaluation shows that only a subset of biological strategies achieves strong alignment with swarm intelligence principles when mapped to UAV communication needs. The honeybee waggle dance and deer tail-flagging demonstrate ✓ strong translatability because they combine clarity, symbolic simplicity, and group-level interoperability, while the peacock display offers ✓ strong role distinction through persistent symbolic cues. In contrast, strategies such as pheromone trails, pronking, and ant task allocation exhibit × weak or ~ partial alignment due to limited visual or maneuver-based mapping, either because they depend on conditions not transferable to UAV contexts or require recurrent information sampling. By structuring this comparison through the checklist rubric, these bio-inspired communication strategies provide a solid foundation for UAV visual signaling.

### 3.2. Reference Architecture

This section outlines a reference architecture for the proposed bio-agentic visual communication system, which enables UAV swarms to exchange mission-critical information visually in RF-denied environments. As shown in Figure 2, each UAV has a perception module, a bio-strategy database, and a GenAI layer.

The perception module receives inputs about threats, obstacles, and neighboring UAV behaviors. In a full implementation, it would include onboard cameras, image classifiers, and spatial reasoning algorithms to estimate angles to threats, detect obstacles, and track the movement and LED states of nearby UAVs. While critical for real-world deployment, the perception module is abstracted in this work to focus on the core functionality and novelties proposed in this approach.

The GenAI layer consists of a Retrieval-Augmented Generation (RAG) model and a Large Language Model (LLM). The RAG component retrieves relevant biological communication strategies, such as bee waggle dances, white-tailed deer flagging, or peacock feather displays, based on the UAV’s current observations. These strategies serve as heuristics for signaling spatial direction, alerts, and roles. The LLM then processes a structured prompt built from the UAV’s current perceptions and the retrieved templates to generate a corresponding visual response.

The perception inputs are tokenized into a maneuver and LED-based sequence grounded in the UAV’s configuration space, which includes position, orientation, and LED states over time. For example, UAV_1_ may detect a threat and signal its direction using a bee waggle dance, while UAV_3_ may perceive an obstacle and emit a deer tail-flagging LED pattern to indicate lateral proximity. Thus, UAV_2_ in the center, observing both signals, uses its own GenAI layer to retrieve matching strategies, interpret the corresponding meanings of the signals from its neighbors, and generate an integrated response. This approach allows visual information to flow through the swarm without the need for RF communication.

Note that these UAVs are assumed to be homogeneous in terms of hardware and software in this proof of concept, yet because the GenAI Layer enables UAV agents to learn and adapt independently based on their perceptions, they may develop their own role specializations, as implied in the figure, thus becoming heterogeneous in nature. This is linked to local perception limitations that exist in UAVs and in nature, therefore it is likely that a central UAV might focus on perceiving its neighbors’ signals, and then it may decide to take on a different role, such as lead interceptor, chaser, or ambusher.

### 3.3. Bio-Strategies in the UAV’s Visual Configuration Space

Three strategies: the honeybee waggle dance, the peacock feather display, and the white-tailed deer flagging, were selected as the foundation for the swarm’s visual communication language. Based on the literature review and morphological evaluation, these are biologically grounded, UAV transferable signals for conveying direction, role, and warnings. They provide a minimal yet representative proof of concept for initial model development, with the next step being to encode them into the UAV’s visual configuration space to convert them into interpretable spatiotemporal behaviors for preconditioning the LLM.

The configuration space of a UAV is the set of all possible positions and orientations it can assume in three-dimensional space. This space is formally represented as $R^{3}\;\times \;SO(3)$, where $R^{3}$ captures translational motion (x, y, z) and SO(3) represents rotational orientation (roll, pitch, yaw) as a rotation matrix. This 6-degree-of-freedom (6-DOF) model allows for full rigid-body motion in 3D space [74]. The visual configuration space of a UAV is a modification to the traditional configuration space that defines the UAV’s complete visual state at a given time, including the same spatial position and orientation components, with the addition of variables for LED states:(1)$$\Pi_{i}(t_{j})=[t_{j},{ID}_{i},x_{i}(t_{j}),y_{i}(t_{j}),z_{i}(t_{j}),\phi_{i}(t_{j}),\theta_{i}(t_{j}),\psi_{i}(t_{j}),L_{i1}(t_{j}),L_{i2}(t_{j}),L_{i3}(t_{j})]$$

The vector $\left(\Pi_{i}(t_{j})\in R^{11}\right)$ captures the global timestamp $t_{j}$ and UAV i’s unique identification (${ID}_{i}$) number, which are included to support the LLM’s reasoning. It also contains its 3D spatial coordinates $\left(x_{i}(t_{j}),y_{i}(t_{j}),z_{i}(t_{j})\right)$, orientation expressed in roll-pitch-yaw angles $\left(\phi_{i}(t_{j}),\theta_{i}(t_{j}),\psi_{i}(t_{j})\right)$, and the instantaneous status of the LEDs $\left(L_{i1}(t_{j}),L_{i2}(t_{j}),L_{i3}(t_{j})\right)$, where each LED can be off or display a primary color from the discrete set {0,R,G,B}, are included. For this proof of concept, we assume only three LEDs per UAV and a limited set of primary colors to simplify the representation while still enabling clear, structured visual signaling sufficient for initial model development and evaluation. For example, a UAV may visually observe its neighbor perform a signaling pattern captured as an input matrix over several time steps:(2)$$\Pi_{i}(t_{1\to n})=[\begin{array}{ccccccccccc} t_{1} & ID_{i} & x_{i}(t_{1}) & y_{i}(t_{1}) & z_{i}(t_{1}) & \phi_{i}(t_{1}) & \theta_{i}(t_{1}) & \psi_{i}(t_{1}) & L_{i1}(t_{1}) & L_{i2}(t_{1}) & L_{1i}(t_{1})\\ t_{2} & ID_{i} & x_{i}(t_{2}) & y_{i}(t_{2}) & z_{i}(t_{2}) & \phi_{i}(t_{2}) & \theta_{i}(t_{2}) & \psi_{i}(t_{2}) & L_{i1}(t_{2}) & L_{i2}(t_{2}) & L_{1i}(t_{2})\\ \vdots & \vdots & \vdots & \vdots & \vdots & \vdots & \vdots & \vdots & \vdots & \vdots & \vdots \\ t_{n} & ID_{i} & x_{i}(t_{n}) & y_{i}(t_{n}) & z_{i}(t_{n}) & \phi_{i}(t_{n}) & \theta_{i}(t_{n}) & \psi_{i}(t_{n}) & L_{i1}(t_{n}) & L_{i2}(t_{n}) & L_{1i}(t_{n}) \end{array}]$$

The bee waggle dance is implemented as a forward, oscillating trajectory aligned with a fixed yaw angle from the x axis, with pitch variation and green LEDs indicating threat direction. The peacock display uses a vertical rise, hover, and descent with a pulsing blue center LED to signal interceptor or ambusher role. Deer tail-flagging is encoded as a stationary hover with red LED sweeps on the side facing the obstacle to signal alerts. The LED coloration assigns one primary color per strategy to allow the LLM to more clearly differentiate them. Figure 3 shows these visualizations of the database in 3D space over time.

These LED patterns were chosen to allow a UAV to potentially combine and communicate multiple patterns simultaneously. For example, if the center UAV perceived one neighbor using the bee waggle dance and another neighbor using white-tailed deer flagging, the response from the center UAV may emerge with a green LED on the right recognizing the threat as it changes its direction and bobs up and down, a blue LED in the center indicating its interception role as it changes its height, and a red LED on the right indicating its recognition of the obstacle. Furthermore, it is assumed for this proof of concept that all three LEDs are visible, with the understanding that future development would require analyzing the directionality and partial observability of the LEDs from UAV-to-UAV. Also, the x, y, and z dimensional units are in meters, though these trajectories may be adjusted within the defined space to be larger or smaller based on the achievable velocities of a specific UAV model selected in the future.

Thus, these behaviors form a baseline visual language of agent interaction protocols, vectorized into a bio-strategies database and rendered as a time-series input for LLM preconditioning via RAG. Defined through trajectories, orientations, and LED patterns, they create a structured vocabulary that grounds model outputs in biologically plausible visual communication.

### 3.4. Test Case Design and Evaluation

To evaluate the proposed bio-agentic communication system, five test cases simulate biologically inspired signaling propagated through the UAV swarm. Each one tests the ability of one UAV to perceive visual behaviors, retrieve a relevant bio-strategy, and respond through movement and LED patterns accordingly. Specifically, test cases evaluate the use of each perception input separately, then the outputs are propagated through the swarm, and finally, a test where both inputs are received:▪Test Case 1: UAV_1_ performs a bee waggle to indicate threat direction. UAV_2_ observes this and responds.▪Test Case 2: UAV_3_ then responds to UAV_2_’s response from Test Case 1.▪Test Case 3: UAV_3_ performs a deer flagging pattern to signal a nearby threat, prompting UAV_2_ to respond.▪Test Case 4: UAV_1_ then responds to UAV_2_’s response from Test Case 3.▪Test Case 5: UAV_2_ perceives both UAV_1_’s waggle dancing and UAV_3_’s tail flagging and generates a single, composite visual response.

To evaluate agent communication effectiveness and determine whether UAV responses are clear and expressive, we adapt the Spike-Triggered Covariance (STC) method as a principled statistical tool. Though originally developed for identifying sensory feature tuning in neurons, STC offers a compelling framework in this context because it quantifies how specific input features systematically influence output behavior [75]. In this case, each UAV’s perception matrix over time (e.g., from observing a waggle dance or tail flag) can be interpreted as a structured input stimulus, while the responding UAV’s trajectory and LED behavior serve as its output. STC is particularly suited because it identifies which features in the observed behavior are preferentially amplified, preserved, or suppressed in the response.

The method compares the input covariance matrix to the input-output cross-covariance written as follows:(3)$$C_{\mathrm{in}}=\frac{1}{n-1}{\tilde{X}}^{⊤}\tilde{X}$$(4)$$C_{\mathrm{cross}}=\frac{1}{n-1}{\tilde{X}}^{⊤}\tilde{Y}$$
where $\tilde{X}\in R^{n\times m}$ represents the mean-centered time-series input matrix of observed features (e.g., $\Pi_{i}\left(t_{1\to n}\right)$, but only focusing on the m maneuver and LED dimensions), and $\tilde{Y}\in R^{n\times m}$ represents the mean-centered response matrix. The difference $\Delta C=C_{\mathrm{cross}}-C_{\mathrm{in}}$, where $\Delta C\in R^{m\times m}$, identifies dimensions where the system’s output systematically reflects its perception, rather than reacting emergently.

A strong diagonal structure in ΔC would indicate within-modality fidelity. For example, If UAV_1_ performs a waggle dance using oscillatory yaw movement to indicate a direction, and UAV_2_ responds with a similar yaw-based movement, this demonstrates high within-modality fidelity. This matters, because it would indicate that the communication system is preserving the channel-specific semantics of a signal, supporting clarity and interpretable transmission across the swarm. Large off-diagonal components might suggest cross-modal translation. For example, if UAV_3_ observes a tail-flagging pattern from UAV_2_ via a raised taillight (an LED cue), but responds with an evasive vertical climb (a motion cue), that would be cross-modal translation. Eigen-decomposition of ΔC reveals the most influential input patterns that shape output behavior, allowing us to measure which bio-strategic elements are preserved:(5)$$\Delta C=Q\Lambda Q^{⊤}$$
where Q contains the eigenvectors and $\Lambda =diag(\Lambda_{1},\Lambda_{2},\ldots ,\Lambda_{m})$ contains eigenvalues along its diagonal. Large eigenvalues indicate strong preservation or amplification of specific input features, supporting expressiveness, while eigenvectors concentrated in single modalities (e.g., motion or LED alone) reflect clarity through within-modality fidelity. In contrast, mixed-modal eigenvectors suggest cross-modal translation, which may enhance expressiveness but reduce clarity if the mapping is ambiguous.

This analysis helps uncover whether UAV responses encode intended signals faithfully and through appropriate channels. This analysis is particularly informative for composite cases like Test Case 5, where UAV_2_ perceives both a bee waggle (UAV_1_) and a deer flag (UAV_3_) and must integrate them into a unified response. Here, we hypothesize that the resulting ΔC will reflect salient features from both inputs. In simpler cases (e.g., Test Case 1 or 3), we expect ΔC to show the strongest values in the covariance matrix along matching channels, validating that the system preserves the symbolic structure of the observed signal.

Although the covariance and eigenvalues provide strong qualitative insights, two quantitative metrics are also defined to allow for a more systematic analysis. The following metrics adapt established principal component analysis (PCA)-based interpretations of covariance structure and eigenvalue variance by treating diagonal concentration as a measure of within-modality fidelity (clarity) and the relative dominance of the largest eigenvalue as an indicator of amplified signal pathways (expressiveness) [76]. Clarity is measured as the diagonal dominance ratio:(6)$$Clarity=\frac{\sum_{m}|\Delta C_{m}|}{\sum_{p,q}|\Delta C_{pq}|}$$
where the numerator sums the magnitude of diagonal entries, capturing signal preserved within the same channel (e.g., pitch input leading to pitch output), and the denominator sums the magnitude of all entries in ΔC (note that p and q represent the row and column indices for ΔC), thus capturing the total structured input to output mapping across all channels. This gives a quantitative determination of how much of the covariance is concentrated along the diagonal. Values closer to 1 indicate strong within-modality fidelity, while lower values indicate cross-modal translation. Expressiveness is measured as the variance explained by the largest eigenvalue:(7)$$Expressiveness=\frac{max(|\Lambda_{m}|)}{\sum_{m}|\Lambda_{m}|}$$
where the numerator is the magnitude of the largest eigenvalue, corresponding to the single most amplified feature pathway, and the denominator is the total variance explained by all eigenvalues. This reflects how dominant one signal pathway can be. Higher values suggest that a feature is strongly amplified and expressed in the response, while lower values indicate weaker expression. To interpret these metrics consistently, we define success thresholds inspired by PCA practice. A trial is considered successful if clarity ≥ 0.70 (indicating strong within-modality fidelity) and expressiveness ≥ 0.50 (indicating amplification of a dominant feature pathway). Trials that satisfy only one of these thresholds are classified as partially successful, while trials that fail both are deemed unsuccessful.

### 3.5. Algorithmic and Mathematical Framework

The algorithm begins with a swarm of UAV agents ($U=\{U_{1},U_{2},\ldots ,U_{i}\}$), a vectorized database of bio-strategies ($D=\{d_{1},d_{2},\ldots ,d_{h}\}$) with corresponding semantic meanings, the GenAI Layer ($M_{\theta }$) consisting of an LLM with integrated RAG, and a sequence of perception event windows ($E=\{e_{1},e_{2},\ldots ,e_{k}\}$). For each UAV ($U_{i}$) at an event window (e), the system generates an output trajectory ($\xi_{i}^{e}$) and LED pattern ($\lambda_{i}^{e}$). The initialization prompt defines the UAV’s role in the swarm, its assumptions, and objectives, grounding the LLM in the symbolic task space before perception begins. This ensures responses remain coherent with swarm roles, as described in the initialization step of the algorithm found in Table 3.

Step 1: Perceive. Each UAV gathers perceptual inputs from its neighbors over an event window. We denote this as:(8)$$\Pi_{i}(t_{1\to n})=\mathrm{Perceive}(U_{i},e)$$
where $\Pi_{i}(t_{1\to n})$ encodes observed trajectories (x, y, z, roll, pitch, yaw) and LED states over time as previously defined. This raw observation stream forms the spatiotemporal grounding of the swarm interaction.

Step 2: Tokenize. Perceptions are transformed into a symbolic sequence suitable for language modeling:(9)$$T_{i}^{e}=\mathrm{Tokenize}(\Pi_{i}(t_{1\to n}))$$
where tokens encode discrete bio-strategy patterns (e.g., waggle oscillation, tail-flagging, or peacock display) into units that are convertible as numerical inputs to the LLM. This step abstracts the continuous motion/LED signals into symbolic units, operationalizing the principle of salience and selective attention.

Step 3: Prompt. Concurrently, the tokenized perception is embedded within a structured natural language input:(10)$$\pi_{i}^{e}=\mathrm{Prompt}(T_{i}^{e})$$
where $\pi_{i}^{e}$ contains the observed tokens, swarm role metadata, and formatting rules. This mirrors the symbolic density principle, compressing rich visual signals into interpretable cues for LLM reasoning.

Step 4: Retrieval. The RAG mechanism queries the database D of biological exemplars. Each entry $d_{h}$ is embedded as $k_{h}=\phi (d_{h})$, while the query embedding is $q_{i}^{e}=\phi (T_{i}^{e})$. Similarity is computed as a normalized dot product between the query vector and exemplar vector:(11)$$s_{ih}^{e}=\frac{q_{i}^{e}\cdot k_{h}}{∥q_{i}^{e}∥∥k_{h}∥}$$
where the top K matches form the retrieval set:(12)$$C_{i}^{e}=TopK_{1}^{h}\left(d_{h},s_{ih}^{e}\right)=RAG(\pi_{i}^{e},D)$$
to ensure that swarm responses are not generated in isolation, but are conditioned on reference biological strategies, grounding novel outputs in meaningful precedents.

Step 5: Reason. The generative model integrates perception and retrieval into a conditional distribution:(13)$$O_{i}^{e}∼M_{\theta }(\pi_{i}^{e}\cup C_{i}^{e})$$
where $O_{i}^{e}$ is a symbolic response sequence. This step corresponds to the LLM aligning context with retrieved exemplars to construct a coherent visual “sentence” of motion and LED behavior. The conditional nature of this generation follows the retrieval–generation paradigm [77] and a general mathematical model of this GenAI Layer is expressed as:(14)$$p(O_{i}^{e}|\pi_{i}^{e})\;=\;\sum_{d_{h}\in D}p(d_{h}|\pi_{i}^{e})\;p_{\theta }(O_{i}^{e}\;|\;\pi_{i}^{e},d_{h})$$
where the first term represents how strongly the retriever judges each biological strategy to match the UAV’s current perception, the second term represents how likely the generator is to produce a particular response given both the perception and a chosen strategy, and the summation combines these possibilities, weighting and blending across all candidate strategies rather than relying on just one.

In practice, this means concatenating the perception prompt with the retrieved exemplars into a single input token sequence, which is then processed through the LLM’s transformer. Each layer of the LLM computes how strongly each token in the input sequence should influence the prediction of the next token, allowing the model to weigh both the UAV’s perception tokens and the retrieved biological exemplars so that the most contextually relevant information influences the next prediction. The model then outputs a probability distribution over the possible maneuver and LED tokens to represent the likelihood that it should be generated next in the UAV response.

Step 6: Respond. Symbolic outputs are mapped to executable UAV actions:(15)$$\left(\xi_{i}^{e},\lambda_{i}^{e})=\Gamma (O_{i}^{e}\right)$$
where $\xi_{i}^{e}$ denotes the motion trajectory and $\lambda_{i}^{e}$ the LED pattern. This ensures interoperability, translating abstract tokens into concrete spatiotemporal signals aligned with swarm dynamics. The tokens are selected from the aforementioned distribution, balancing fidelity to the input and variability for expressiveness. They are appended to the response sequence until a complete trajectory and LED pattern are produced.

Step 7: Execute. Finally, UAV $U_{i}$ enacts the behavior:(16)$$\mathrm{Execute}(U_{i},\xi_{i}^{e},\lambda_{i}^{e})$$
where trajectories and LED patterns are produced according to the initialization specifications, the context received through RAG, the LLMs capabilities, and the perception input communicated to the UAV.

### 3.6. Experimental Setup

After identifying biological communication strategies in Section 3.1 through morphological evaluation, the experimental setup proceeds as follows:▪The first step was to create an environment that runs on a selected GPU and can make API calls to the LLM (e.g., NVIDIA A100 and OpenAI GPT-4o in this case, respectively). This environment manages communication between a UAV agent’s GenAI layer, the vectorized biological strategy database, and the perception inputs (as defined in Section 3.2). The environment is configured to handle structured input/output exchanges and to log all system responses for downstream analysis.▪Biological communication strategies (e.g., bee waggle dances, deer tail-flagging, peacock displays) are represented in as 3D trajectories and LED patterns across 13 discrete timesteps, according to Section 3.3. Each strategy is stored as a vectorized item in the database along with metadata that describes its semantic meaning (threat direction, role signaling, or obstacle location). Embeddings are computed using a pretrained transformer model (e.g., sentence-transformers/all-MiniLM-L6-v2), and the database is indexed in FAISS for fast similarity search. The retriever is configured to return the top-k most relevant matches.▪Test cases are built by creating perception data that resembles, but is not identical to, the stored database strategies. This ensures that the LLM must infer the meaning of new perceptions through reasoning by using the database entries as a starting point rather than something to copy exactly. Each test case defines a specific signaling scenario as defined in Section 3.4 and is executed from the perspective of the UAV receiving the perception input. This UAV compares the perceived sequence from the test case with the database entries, retrieves relevant strategies, and uses that to generate an appropriate response.▪Each UAV is initialized with a structured system prompt that establishes its role, output format, objective, and response rules according to the algorithm defined in Section 3.5. Perception prompts embed the symbolic tokens extracted from observed trajectories and LED states, which are passed to the LLM along with the retrieved strategies. These prompts are submitted via API calls, executed on the GPU, and the responses consist of output trajectories and LED patterns generated by the LLM accordingly.▪The responses are then analyzed post-experiment. Analysis includes computing the covariance matrices and eigenvalues across communication channels based on the perception input and its corresponding response output, and calculating the quantitative metrics for clarity and expressiveness scores as defined in Section 3.4.

## 4. Results

This section presents the results and discussion of five test cases used to evaluate the proposed bio-agentic visual communication system for UAV swarms. Each test case demonstrates how UAVs interpret and respond to biologically inspired visual signals in RF-denied environments. The analysis includes both qualitative behavior outputs using covariance matrices and eigenvalue decompositions, along with quantitative evaluations of expressiveness and clarity. The section concludes with a summary of key findings and outlines directions for future work.

The results of the five test cases demonstrate that this bio-agentic visual communication proof of concept successfully enables signal interpretation, symbolic encoding, and swarm-level propagation across UAV agents. They provide a direct evaluation of the hypotheses introduced in the methods section. Core to this was the expectation that the proposed bio-agentic communication system would enable UAVs to interpret and respond to visual signals in a manner that is both clear and expressive, and that this would manifest as structured differences in the covariance matrix (ΔC). Furthermore, it was anticipated that strong diagonal elements in ΔC would reflect within-modality fidelity, preserving the symbolic structure of the observed signals, while off-diagonal components would suggest cross-modal translation, potentially supporting expressive integration but at the risk of reduced clarity. For each test case, results are presented in a standardized format that includes: (A) UAV perception input, (B) UAV response output, (C) the ΔC heatmap, and (D) the eigenvalue spectrum, providing a comprehensive view of how biomechanics, covariance structure, and eigen-decomposition jointly characterize the communication outcome.

### 4.1. Test Case Results and Discussion

Test Case 1: In the first test case, UAV_1_ performs a bee waggle dance trajectory with green LEDs at a yaw angle of 50· to indicate the threat direction. UAV_2_ observes this behavior, checks its baseline communication reference database containing a bee waggle dance at a yaw angle of 30·, and decides to respond with a similar bee waggle dance trajectory at the correct angle of 50·. However, it converts its center LED to blue, presumably to indicate that its role has changed to interceptor, thereby mixing the bee waggle dance and the peacock feather display into an emergent communication. This response suggests that UAV_2_ correctly interpreted the motion-based signal as directional information and expressed its understanding using a distinct visual modality through LED signaling.

As shown in Figure 4, the covariance and eigenvalue analysis strongly supports the hypothesis of within-modality fidelity. The covariance shows pronounced values involving pitch, which biomechanically drives oscillatory rise-and-fall in the z direction, but the eigen-decomposition projects this variation most strongly into the y channel, where the dominant eigenvalue of −1472.698 appears. This indicates that the system not only preserved the modality of the signal, oscillatory motion through waggle-induced displacement but also amplified it, supporting both clarity and expressiveness.

The preservation of the pitch/z dynamics from the input and their projection into y in the output demonstrates that the semantic content of the waggle dance was successfully encoded and transmitted through motion channels without cross-modal confusion. This suggests that for highly structured signals like directional movement, the system enables effective one-to-one mapping from perception to response. This interpretation is quantitatively confirmed as a clear success by a Clarity score of 0.992 and Expressiveness of 1.000, indicating nearly perfect within-modality fidelity and dominance of the preserved waggle-related motion pathway.

Test Case 2: In the second test case, UAV_3_ interprets this response from UAV_2_ by displaying an ascending trajectory while activating green LEDs on both sides. This change in both motion and LED configuration presumably reflects UAV_3_’s acknowledgment that the swarm has identified a target, noted by its own green LEDs and yaw angle maintenance at 50, and that it understands that another UAV has taken on the role of interceptor, thus keeping its center LEDs off instead of turning them blue if it were to take on the role itself. UAV_3_ may be signaling that it is ready to take on a supporting role for the interception. Biomechanically, this output is dominated by a steady vertical climb (z) with yaw held at 50°, while the side LEDs perhaps indicate readiness rather than a role change.

As illustrated in Figure 5, the covariance shows some structure, but with much lower magnitude. A few off-diagonal elements appear, indicating a blend of z and pitch features, while the largest eigenvalue is −0.191 in the z channel. Interpreting the off-diagonal terms from x and y to pitch indicates a cross-modal mapping where lateral cues from the perceived waggle are converted into pitch adjustments that effect ascent, while the only substantive diagonal term in z reflects weak within-modality fidelity.

These modest values suggest limited amplification and some signal dampening across the propagation chain. While the system retains some expressiveness, the clarity appears reduced, likely due to the transformation or weakening of the original input as it passed through UAV_1_ → UAV_2_ → UAV_3_. The result is a weaker form of within-modality fidelity, with hints of emergent cross-modal associations. This communication is supported as partially successful by an Expressiveness of 1.000 alongside a very low Clarity value of 0.003, further reflecting weak fidelity but some amplification of blended features.

Test Case 3: The third test case evaluates a separate signaling mode, where UAV_3_ performs a straight-line movement with its right LED flashing red, simulating a White tailed Deer’s tail-flagging behavior to indicate an obstacle alert nearby. UAV_2_ mirrors this signal but adds a lateral offset path over time, veering its path away from the obstacle, reflecting its understanding of the alert and maintaining its red LED pattern, effectively relaying the signal forward to its neighbors. Biomechanically, this is an LED-dominant alert behavior, relying on conspicuous symbolic flashing rather than strong maneuvering.

As shown in Figure 6, the covariance is nearly zero throughout, with only small entries in x and y outputs. This indicates that UAV_2_ introduced a slight lateral displacement correlated with forward motion. The eigen-decomposition projects this into a single small negative eigenvalue of −0.037, with all other modes collapsing to zero. Thus, UAV_2_ generated only one weak structured response, without amplification or multimodal coupling.

Given that the deer flag is primarily a visual LED cue rather than a maneuver, this result suggests that the system struggled to interpret or translate this type of signal. The response biomechanically emphasized symbolic LED flashing, but the covariance/eigenvalue analysis showed that the system detected almost no structured correlation, highlighting a gap between intended signaling and the system’s response. This outcome challenges the clarity aspect of the system’s design, revealing a gap in its responsiveness to certain types of visual signaling, especially those that rely purely on symbolic LED behavior rather than motion, which will have to be accounted for in future designs. The calculated Clarity and Expressiveness scores of 0.080 and 1.000, respectively, confirm the presence of only a very weak preserved structure and dominance of a single trivial mode, thereby confirming Test Case 3 as unsuccessful while uncovering valuable lessons learned.

Test Case 4: In Test Case 4, UAV_1_ responds to UAV_2_’s behavior from Test Case 3. Biomechanically, this involved UAV_2_’s red LED relay and slight lateral offset being observed by UAV_1_, which then attempted to reproduce the signal while continuing straight flight. As shown in Figure 7, the covariance shows very small structure, with weak off-diagonal correlations between x and y position features, and the eigenvalues are extremely small, the largest being −0.037 with the rest collapsing to zero, indicating that no dominant response pathway was activated. This could reflect ambiguity in the perceived signal or weak interpretability of UAV_2_’s prior action.

While there is some evidence of responsiveness, it lacks clarity, suggesting that once a signal is either poorly formed or misunderstood by an intermediate agent, it becomes difficult for downstream agents to reconstruct or respond in a more meaningful way. Biomechanically, this was emphasized as a clear symbolic LED relay, but the covariance/eigenvalue results show only a trivial, non-amplified structure. This is reflected in the metrics, where Clarity is only 0.298 but Expressiveness is 1.000, showing a partial success, where the response mathematically collapsed into a single trivial mode but failed to yield a clear within-modality mapping.

Test Case 5: In the final test case, UAV_2_ perceives both the bee waggle dance from UAV_1_ and the deer flag from UAV_3_. It generates a hybrid response by combining the motion patterns associated with directional threat identification signaling, additionally activating the center blue LED to signal its role as interceptor, and activating its right red LED to indicate its recognition of the obstacle alert. This output reflects accurate integration of multiple distinct messages into a coherent visual language, capturing both directional guidance and threat awareness. Overall, the results show that UAVs can detect, interpret, and symbolically respond to visual signals, maintaining meaning while modulating behavior across agents by leveraging this bio-agentic approach. Thus, this proof of concept demonstrates its capability for visual swarm communication in RF-denied environments.

As shown in Figure 8 and Figure 9, Test Case 5 simulates a composite perception by UAV_2_, which observes both a waggle dance from UAV_1_ and a deer tail flag from UAV_3_. The analysis treats these as two separate input comparisons. Biomechanically, UAV_2_ encountered a structured oscillatory waggle from UAV_1_ alongside a symbolic LED flash from UAV_3_, forcing it to reconcile motion-based and purely visual signaling modes. The matrix comparing UAV_1_’s input produces almost no covariance structure, with extremely small eigenvalues, indicating a weak preservation of waggle dynamics relative to Test Case 1. The second matrix, comparing UAV_3_’s input, shows stronger off-diagonal values, especially linking LED inputs to motion outputs, and yields a dominant eigenvalue of −0.867. This indicates that the deer’s symbolic LED flagging was projected into motion space, creating a cross-modal transformation absent in simpler cases.

This supports expressiveness, as UAV_2_ was able to integrate inputs from distinct signaling styles, but it also introduces ambiguity by reducing the within-modality clarity seen in simpler cases. The metrics align with this: the UAV_1_→ UAV_2_ comparison yields a Clarity of 0.018 with an Expressiveness of 0.969, while the UAV_3_→ UAV_2_ comparison yields another low Clarity of 0.090 with Expressiveness of 1.000, confirming strong amplification but poor within-modality fidelity for the mixed signal case as a partially successful test.

The analysis confirms that the system excels at preserving motion-based signals through within-modality fidelity in simple one-to-one exchanges. In more complex or mixed-modality scenarios, the system shows signs of cross-modal translation, which can increase expressiveness but may reduce clarity. As shown in Table 4, expressiveness scores remain consistently high because responses tend to collapse into a single dominant eigenmode, whereas clarity scores are often lower when this mode does not align with the intended input channel.

This tradeoff highlights the importance of further tuning how UAVs weigh and combine visual inputs, especially when multiple cues arrive simultaneously or when the system must bridge symbolic biological signaling types. In addition, the near-zero covariance and trivial eigenvalues associated with LED-driven cases emphasize that investigating more dynamic and semantically rich LED patterns will be key in future work to evaluate their potential for clear and expressive visual signaling. Overall, the five test cases show that the proposed system enables UAVs to act as reasoning agents who interpret visual signals. Fidelity is strongest for structured motion-based cues, while symbolic LED-based signals remain underdeveloped, and mixed-modality cases highlight the tradeoff between clarity and expressiveness. These results validate the feasibility of bio-agentic swarm communication, while pointing to specific challenges in extending symbolic LED channels and balancing multimodal integration.

### 4.2. Conclusions and Future Work

The results confirm the feasibility of our bio-agentic approach while highlighting challenges for real-world deployment. Test Cases 1–2 showed that motion-based signals were well preserved and propagated. However, environmental disturbances such as turbulence, atmospheric shimmer, and partial LED occlusion can distort both maneuver perception and light-based cues, raising the need for trials with physical UAVs. The limited fidelity of LED-only signaling in Test Cases 3–4 shows that static light cues are insufficient, motivating richer symbolic languages using dynamic patterns, such as Morse-like pulsing, frequency modulation, or multi-color codes to improve clarity. Scaling also requires rethinking swarm composition. Homogeneous UAVs show proof-of-concept feasibility, but role-specialized heterogeneity, where agents focus on perception, signaling, or decision-making, could support division of labor at scale. The mixed outcomes of Test Case 5, where multiple signals increased expressiveness but reduced clarity, underscore the need for improved signaling strategies, possibly through role specialization.

In comparison with rule-based optical signaling systems, which offer simplicity and predictability yet lack adaptability in contested settings, our approach allows context-aware interpretation and emergent behaviors. The tradeoff of more computational resources, model size, and safety validation indicates the need to benchmark against rule-based baselines. While GPT-4o proved effective for proof-of-concept testing, such models are impractical for edge deployment. Future work will investigate fine-tuning LLMs for UAV-specific communication tasks so they can run locally on embedded GPUs. Finally, adversarial environments pose the risk of spoofing or malicious signal injection. A future-proof system must incorporate responsible AI, such as redundancy and anomaly detection, to mitigate risks and ensure safe, trustworthy operation in defense or disaster response scenarios.

## Figures and Tables

**Figure 1 biomimetics-10-00605-f001:**
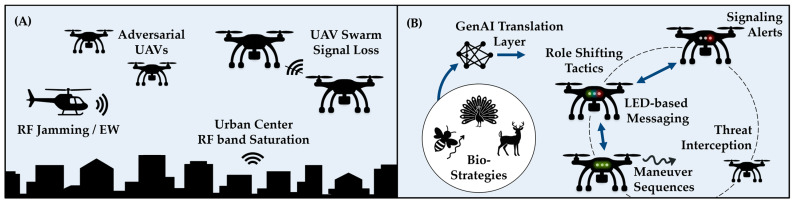
(**A**) RF-denied environments disrupting threat interception capability. (**B**) Bio-Agentic visual communication enabling threat interception through LED and maneuver-based signaling. Note that the blue arrows illustrate how the Bio-Strategies precondition the GenAI Translation Layer and enable the UAVs to propagate visual signals bidirectionally.

**Figure 2 biomimetics-10-00605-f002:**
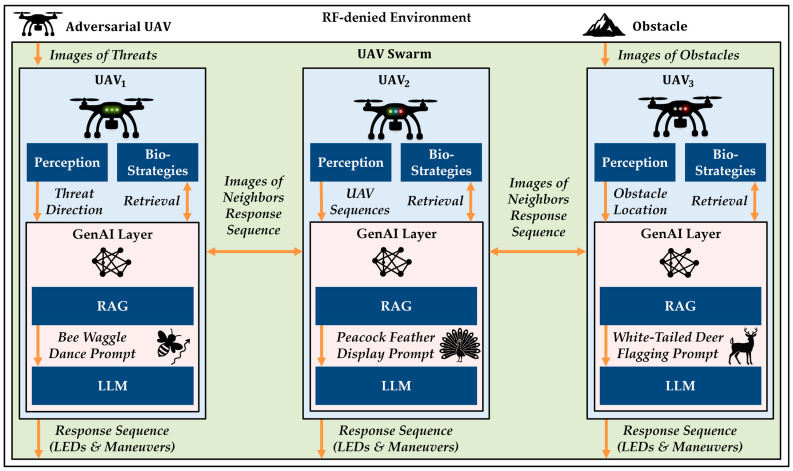
Reference architecture for visual swarm communication in RF-denied environments. Each UAV interprets visual inputs by retrieving bio-strategies and generating a response. Note that the orange arrows illustrate the flow of information from a UAV receiving perception information about adversaries, obstacles, or each other, to sending the perceptions to the GenAI Layer, querying and retrieving Bio-Strategies, sending the resulting prompt to the LLM, and finally producing a response sequence which becomes perceptible in the environment. The environment in white encompasses everything in the figure, while the UAV swarm in green contains the UAV swarm agents, each in light blue. The GenAI Layer is in orange for each agent and consists of the LLM and RAG model. Dark blue boxes represent modular components that each agent utilizes.

**Figure 3 biomimetics-10-00605-f003:**
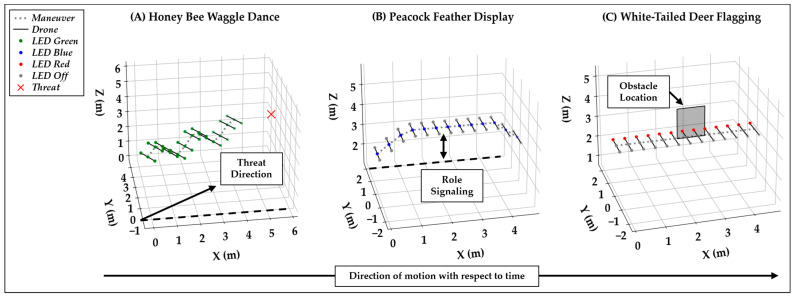
Bio-inspired visual signals encoded in configuration space: (**A**) bee waggle dance for a threat at 30, (**B**) peacock display for role signaling, (**C**) deer tail-flagging for an obstacle on the left.

**Figure 4 biomimetics-10-00605-f004:**
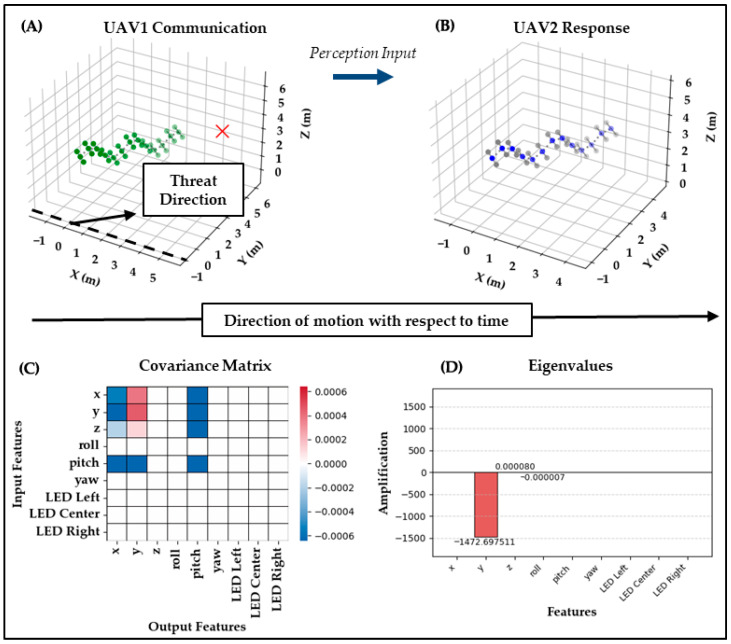
Test Case 1: UAV1 → UAV2. (**A**) UAV1 performs a bee-inspired waggle dance at 50° yaw with green LEDs. (**B**) UAV2 responds with a waggle at the correct angle, but switches its center LED to blue to indicate interceptor role. (**C**) The covariance ΔC shows pronounced values involving pitch, which biomechanically drive oscillatory rise-and-fall in z. (**D**) The eigenvalues project this variation most strongly into y, producing a dominant eigenvalue of −1472.698. Clarity = 0.992 and Expressiveness = 1.000 confirm nearly perfect within-modality fidelity and amplified waggle-related dynamics.

**Figure 5 biomimetics-10-00605-f005:**
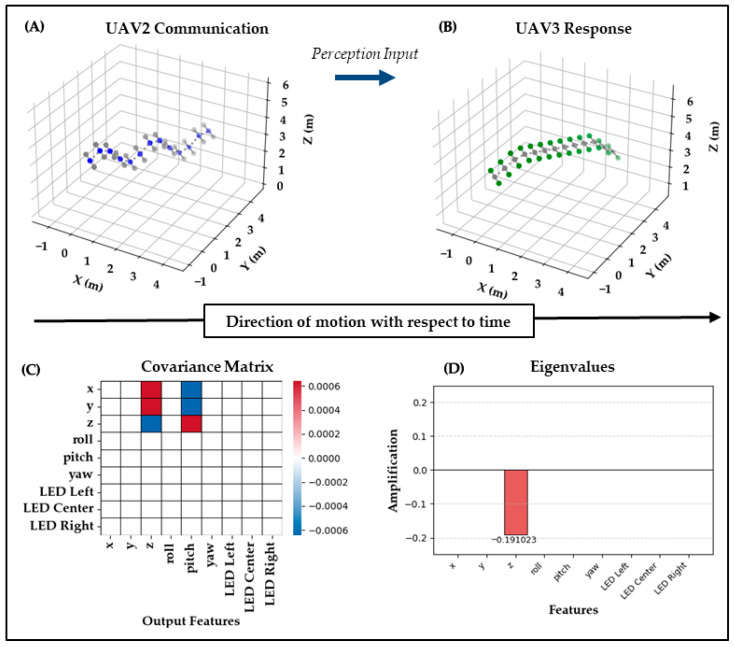
Test Case 2: UAV2 → UAV3. (**A**) UAV2’s waggle relay is perceived by UAV3. (**B**) UAV3 responds with an ascending trajectory, green side LEDs, and yaw maintained at 50°, signaling readiness for support. (**C**) The covariance ΔC shows modest off-diagonal terms linking x and y to pitch, suggesting cross-modal mapping into ascent. (**D**) The eigenvalues yield a largest value of −0.191 in z, confirming weak preservation of waggle dynamics. Clarity = 0.003 and Expressiveness = 1.000 indicate diminished fidelity but preserved amplification.

**Figure 6 biomimetics-10-00605-f006:**
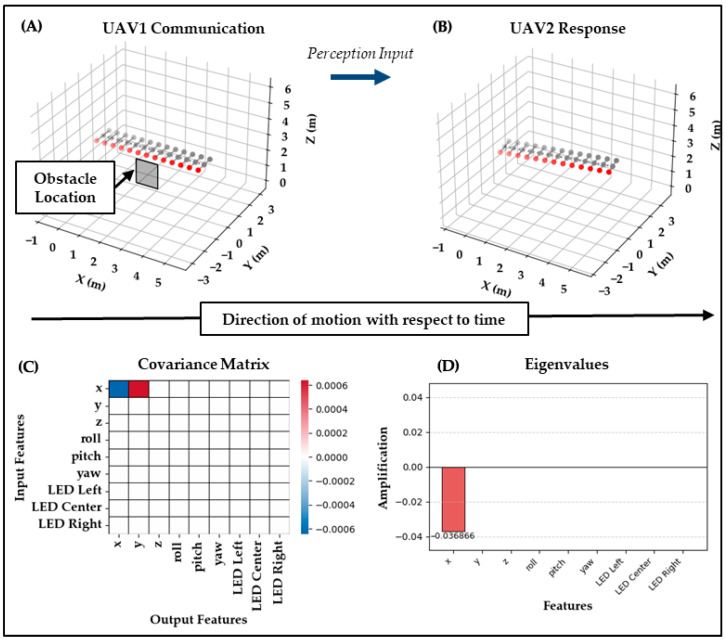
Test Case 3: UAV3 → UAV2. (**A**) UAV3 performs a deer-inspired tail flag with flashing red right LED. (**B**) UAV2 responds with only a slight lateral offset while maintaining its red LED, representing a weak relay of the alert. (**C**) The covariance ΔC is nearly zero, with only small entries in x and y, reflecting minimal structured response. (**D**) The eigenvalues show a single small value of −0.037 with all others collapsing to zero. Clarity = 0.080 and Expressiveness = 1.000 indicate one trivial preserved mode, confirming unsuccessful LED-based translation.

**Figure 7 biomimetics-10-00605-f007:**
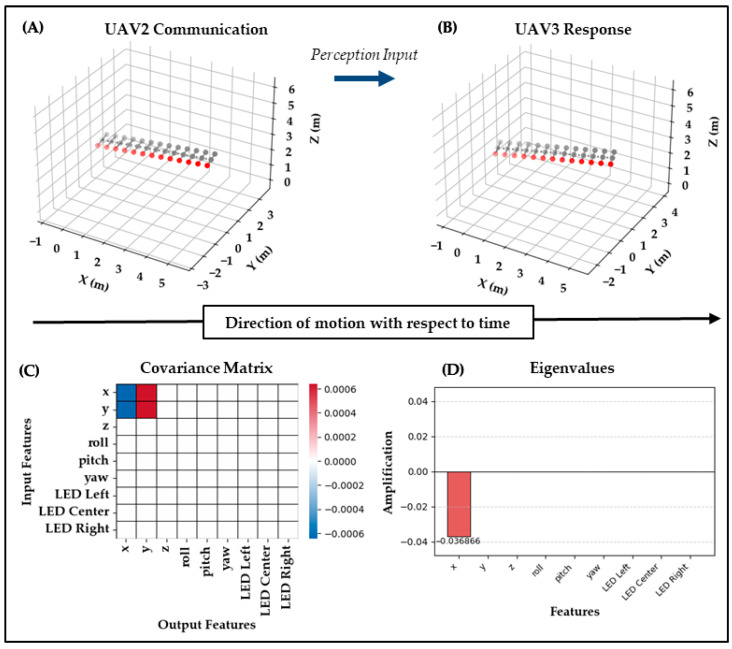
Test Case 4: UAV2 → UAV1. (**A**) UAV2 relays the deer tail flag with red LEDs and slight lateral offset. (**B**) UAV1 attempts to reproduce the signal with straight flight and red LEDs. (**C**) The covariance ΔC contains only very small structure and weak off-diagonal correlations in position. (**D**) The eigenvalues show the largest value as −0.037 with the rest collapsing to zero. Clarity = 0.298 and Expressiveness = 1.000 confirm responsiveness collapsed into a single trivial mode with weak fidelity.

**Figure 8 biomimetics-10-00605-f008:**
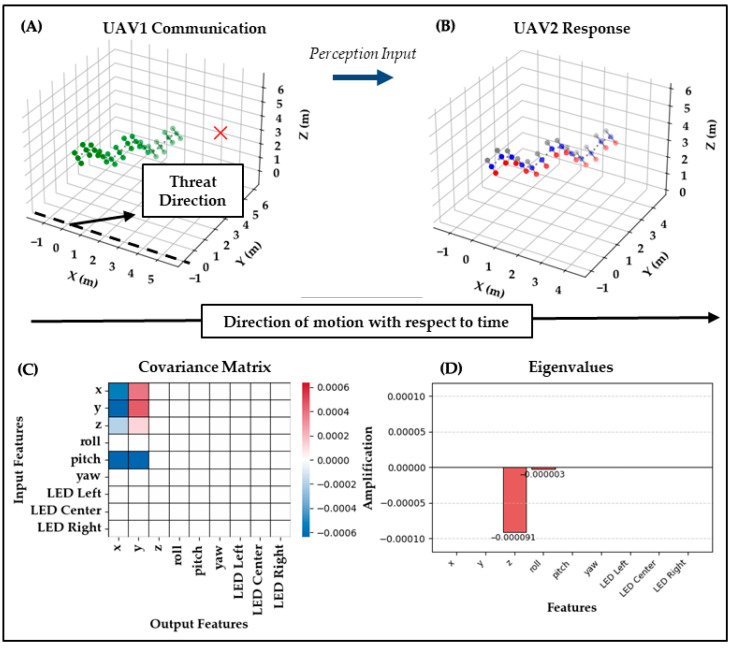
Test Case 5a: UAV1 → UAV2. (**A**) UAV1 performs a waggle dance with oscillatory motion and green LEDs. (**B**) UAV2 produces a hybrid response, retaining waggle motion while activating both blue center and red right LEDs to indicate interceptor role and obstacle recognition. (**C**) The covariance ΔC shows almost no preserved structure compared to Test Case 1. (**D**) The eigenvalues have extremely small magnitudes (largest ≈ −9.1 × 10^−5^). Clarity = 0.018 and Expressiveness = 0.969 indicate weak waggle fidelity despite symbolic integration.

**Figure 9 biomimetics-10-00605-f009:**
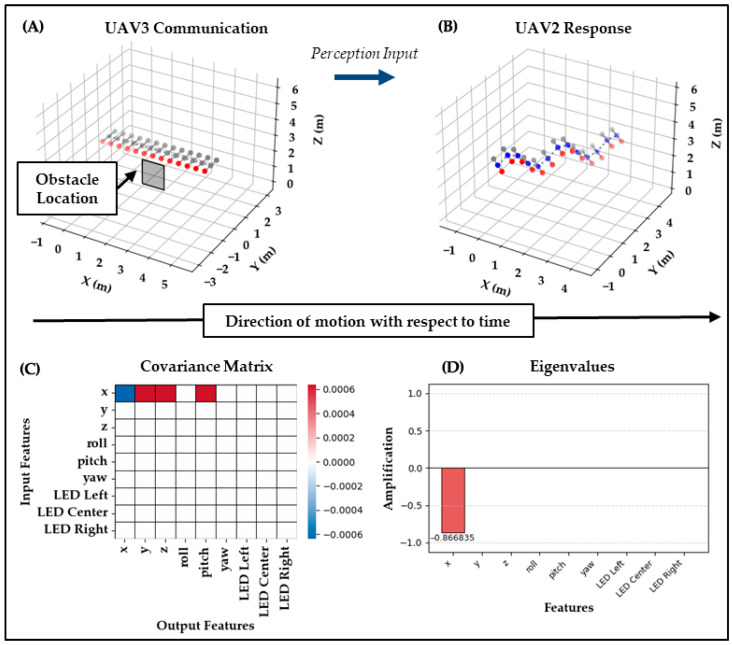
Test Case 5b: UAV3 → UAV2. (**A**) UAV3 performs deer-inspired tail flagging with flashing red LEDs. (**B**) UAV2 integrates this symbolic input into its hybrid waggle context, producing motion adjustments alongside LEDs. (**C**) The covariance ΔC shows stronger off-diagonal values, particularly mapping LED input into motion outputs. (**D**) The eigenvalues yield a dominant value of −0.867, confirming strong amplification of the LED-to-motion transformation. Clarity = 0.090 and Expressiveness = 1.000 reflect cross-modal expressiveness but poor within-modality fidelity.

**Table 1 biomimetics-10-00605-t001:** A communication synthesis: bridging swarm intelligence with biological analogues.

Communication	Swarm Intelligence	Biological Analogues
Target Location: UAVs encode spatial objectives through motion or visual cues.	[15] GenAI supports spatial signaling via flight patterns. [16] FlockGPT maps linguistic goals to directional trajectories. [17] Gen-Swarms coordinates flight paths for group cohesion.	[18] Honeybee waggle dances encode direction and distance. [19] Ant pheromone trails guide movement. [20] Archerfish use line-of-sight targeting.
Obstacle Alerts: UAVs relay hazards through symbolic signals or behaviors.	[21] Agents evolve threat alerts under partial observability. [22] Robust learned protocols amid environmental variability. [23] Communication adaptation in adversarial environments.	[24] Tail and rump displays signal threats to group members. [25] Bat echolocation doubles as obstacle alerts. [26] Gazelle pronking increases visibility and signals danger.
Role or Status: Roles are conveyed through trajectory shifts or LED cues for task alignment.	[27] MARL and federated learning enable local role adoption. [28] LLMs provide context for dynamic role switching. [29] Decentralized messaging supports autonomous tasking.	[30] Lions use movement roles in ambush coordination. [31] Peacocks display visual status cues. [32] Ants assign roles via local task demands.
Proximity: UAVs coordinate to avoid contact using decentralized feedback.	[33] Shared trajectories support real-time avoidance. [34] AI enables predictive spacing and coordination. [35] MARL supports peer interaction formation integrity.	[36] Starling flocks use local alignment to prevent collisions. [37] Geese maintain spacing in energy-efficient formations. [38] Yellowtail fish adjust spacing using local sensing.
Reformation: Swarm cohesion is restored using visual motion cues.	[39] Symbolic UAV motions guide regrouping. [40] Digital trail emulation aids passive reformation. [41] MARL enables restoration of shared intent.	[42] Sardines rapidly reform after predator disruption. [43] Bird-like UAVs use vision to regain formation. [44] Starlings adapt shapes mid-flight to restore cohesion.
Mission Progress: Task status is expressed via motion or light indicators.	[45] Formation and role shifts reflect mission phases. [46] Trajectory morphing encodes semantic outcomes. [47] Visual cues signal progress without central control.	[48] Elephants use trunk gestures to indicate intent. [49] Bowerbirds escalate displays to show readiness. [50] Tree frogs change pulse rates to mark progress.
Redundancy: Distributed signals ensure robustness under interference.	[51] Decentralized alerts provide overlapping coverage. [52] Shared observations build swarm-level awareness. [53] Trajectory cues enable local early warnings.	[54] Ants reinforce pheromone trails with repeated signals. [55] Fireflies flash intermittently for redundancy. [56] Birds deploy multiple sentinels for layered detection.
Intent Prediction: Agents forecast peer actions for anticipatory coordination.	[57] Agents reallocate tasks and reroute after failures. [58] MARL anticipates adversarial behavior. [59] Trajectory prediction supports seamless alignment.	[60] Wolves anticipate escape paths to coordinate encirclement. [61] Orcas block prey escape via synchronized movement. [62] Baboons predict neighbor movement for cohesion.

**Table 2 biomimetics-10-00605-t002:** A morphological identification and evaluation of bio-strategy translatability to UAVs.

Communication	Encoding Precision and Role Cue Clarity	Maneuver and Display Map-Ability	Symbolic Simplicity	Interoperability for Group Response	Best Fit
Target Location:[65] Honeybee Waggle Dance	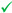 Strong (encodes direction and distance)	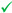 Strong (oscillatory path maps to maneuvers)	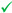 Strong (repeatable symbolic code)	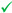 Strong (easily propagated)	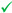 Yes
Target Location:[66] Ant Pheromone Trail	 Weak (low directionality)	 Weak (not motion-based)	 Weak (no visual analog)	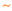 Partial (context-dependent reinforcement)	 No
Target Location:[67] Archerfish Aiming	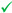 Strong (precise targeting)	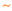 Partial (requires visible reference)	 Weak (not generalizable)	 Weak (limited group propagation)	 No
Obstacle Alerts:[68] White Tail Deer Flagging	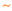 Partial (binary clarity, limited detail)	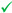 Strong (direct LED translation)	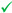 Strong (simple and unambiguous)	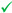 Strong (rapid swarm propagation)	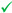 Yes
Obstacle Alerts:[69] Bat Echolocation	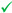 Strong (precise individual navigation)	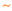 Weak (not visualizable)	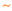 Weak (complex, non-symbolic)	 Weak (not swarm-wide)	 No
Obstacle Alerts:[70] Gazelle Pronking	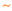 Partial (requires highly specific interpretation of movement)	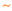 Partial (requires verticality)	 Weak (energy-intensive, not symbolic)	 Weak (low group propagation)	 No
Role or Status:[71] Peacock Display	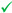 Partial (quality signal, not spatial)	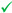 Strong (LED pattern adaptation)	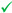 Strong (distinct, symbolic, persistent)	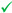 Strong (interpretable at a glance)	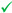 Yes
Role or Status:[72] Lion Ambush Coordination	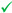 Strong (clear tactical roles)	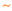 Partial (role inferred, not explicit)	 Weak (not symbolic)	 Weak (coordination without persistent cues)	 No
Role or Status:[73] Ant Task Allocation	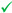 Strong (flexible task shifts)	 Weak (contact-based, not visual)	 Weak (subtle, hard to signal via LEDs)	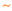 Partial (requires dense encounters)	 No

(✓ = strong alignment, ~ = partial alignment, × = weak alignment).

**Table 3 biomimetics-10-00605-t003:** An algorithm for bio-agentic visual communication.

Steps	Algorithm
Inputs	▪ U = {U_1_, U_2_, …, Uₙ}: UAV swarm agents ▪ D: Vectorized database of bio-strategy exemplars and semantic meanings ▪ $M_{\theta }$: GenAI Layer consisting of an LLM with integrated RAG capability ▪ E = {e_1_, e_2_, …, eₖ}: Sequence of perception event windows
Outputs	A symbolic response parsed into executable visual behaviors for each UAV: ▪ Trajectory ξᵢᵉ ▪ LED pattern λᵢᵉ
Initialization	For each UAV Uᵢ, Pass the following structured initialization prompt to M to establish grounding: ROLE: You are UAV Uᵢ, in a swarm of three. State Format: ▪ Position: x, y, z ▪ Orientation: roll, pitch, yaw ▪ LEDs: Left, Center, Right ∈ {0, R, G, B} Objective: Interpret signals from neighbors using database references. Determine the appropriate, logically consistent response. Your response may combine multiple behaviors (e.g., role + threat direction + obstacle side). Output the result from t = 1 to 13, with columns: ID, t, x, y, z, roll, pitch, yaw, LED Left, Center and Right. Response Strategy: ▪ Combine reference patterns when appropriate. ▪ Use orientation and LEDs to reflect specific meanings. ▪ New composite responses are allowed if plausible. Instructions: ▪ Wait for perception data.
Perception Events	For each perception event e ∈ ε and For UAV Uᵢ ∈ U Step 1: Perceive $\Pi_{i}(t_{1\to n})$← Perceive(Uᵢ, e): Gather peer states over time (positions, orientations, LEDs) Step 2: Tokenize Tᵢᵉ ← Tokenize($\Pi_{i}(t_{1\to n})$): Convert perceptions into symbolic tokens representing visual behaviors Step 3: Prompt $\pi_{i}^{e}$← Prompt(Tᵢᵉ): Provide the LLM with a natural-language prompt of tokens Step 4: Retrieval Cᵢᵉ ← RAG($\pi_{i}^{e}$, D): Retrieve top-matching strategies from the bio-strategy vector database Step 5: Reason Oᵢᵉ ← $M_{\theta }$($\pi_{i}^{e}$∪ Cᵢᵉ): LLM uses prompt and retrieved context to generate symbolic response Step 6: Respond (ξᵢᵉ, λᵢᵉ) ← Γ(Oᵢᵉ): Parse output into a flight trajectory and LED pattern Step 7: Execute Execute(Uᵢ, ξᵢᵉ, λᵢᵉ): Deploy the visually encoded response physically

**Table 4 biomimetics-10-00605-t004:** Resulting summary of expressiveness, clarity, and outcomes across the test cases.

Test Case	Clarity	Expressiveness	Outcome
Test 1UAV1 → UAV2	0.992	1.000	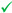 Clear success
Test 2UAV2 → UAV3	0.003	1.000	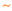 Partial success (expressive but not clear)
Test 3UAV3 → UAV2	0.080	1.000	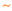 Partial success (expressive but not clear)
Test 4UAV2 → UAV1	0.298	1.000	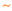 Partial success (expressive but not clear)
Test 5UAV1 → UAV2	0.018	0.969	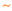 Partial success (expressive but not clear)
Test 5UAV3 → UAV2	0.090	1.000	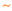 Partial success (expressive but not clear)

## Data Availability

The original dataset used as inputs in the study are openly available: https://github.com/bryanstarbuck314/A-Concept-for-Bio-Agentic-Visual-Communication-Bridging-Swarm-Intelligence-with-Biological-Analogues (accessed on 5 September 2025).

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
