# Peer review of "A Concept for Bio-Agentic Visual Communication: Bridging Swarm Intelligence with Biological Analogues"

_biomimetics, 2025, doi:10.3390/biomimetics10090605_

Round 1

Reviewer 1 Report

Comments and Suggestions for Authors

This paper innovatively combines biological swarm communication strategies with generative AI to construct a UAV visual communication framework for RF-denied environments, and verifies its feasibility through five biologically inspired test cases. The topic is cutting-edge and demonstrates a high degree of interdisciplinary integration. However, the paper still requires improvement in methodological details, logical organization, terminology usage, and data analysis.

Specific Comments

Question 1: The connection between “However, most existing visual systems...” and “Biological systems provide a richer model.” is not sufficiently cohesive, resulting in a logical gap. Adding a transitional sentence that explicitly explains how biological systems address the “static limitations” mentioned earlier would significantly improve the coherence of the argument.

Question 2: The terms ‘interpretable’ and ‘expressive’ each recur multiple times throughout the manuscript, particularly in the abstract and introduction, resulting in redundant phrasing. Recommend streamlining the language to improve readability.

Question 3: All tables are currently presented as images. Please convert them into table format for clearer presentation and easier reference.

Question 4: Reference [14] is listed but never cited in the text. Moreover, some biological examples have inconsistent reference numbering between the literature review and the main text. Please review the entire manuscript to ensure consistent citation numbering, complete coverage of all references, and sort the bibliography by publication year or alphabetically.

Question 5: Although this article employ STC to assess “fidelity” and “cross-modal translation,” but it do not provide quantitative definitions for metrics such as “clarity” or “expressiveness.” Please specify, in the Methods section, the quantitative thresholds used in each test (e.g., signal-to-noise ratio, error rate) or include baseline comparison data.

Question 6: The heatmap in Figure 5 lacks a numerical color scale, making it difficult to interpret the intensity of values. Please add a color bar with clearly labeled tick marks to indicate the corresponding numerical values.

Question 7: In Section 3.1 (“Selected Principles of Communication”), this study introduce three criteria--clarity, symbolic density, and encodability, but do not describe how they are calculated or scored. Please provide the specific formulas or scoring rubric used for each metric.

Question 8: The text refers to “the complete algorithm in Table 3,” but Table 3 is not provided. Readers cannot verify the pseudocode workflow. Please embed the algorithm steps directly in the manuscript.

Comments on the Quality of English Language

A minor to moderate language polishing is necessary.

Reviewer 2 Report

Comments and Suggestions for Authors

Overall, this paper is very interesting. Unfortunately, there are several issues that seems not clear so that it should be improved.

  1. The experiment setup is not clear so that it is difficult to reproduce.
  2. The conclusion is too long so that it should be rewritten.
  3. UAV movement can be seen as intelligent agent/multi agent system. It would be better if authors also makes linkage with agent concept.
  4. The mathematical formulation/concept is too few or insufficient.
  5. Authors should provide the general mathematical model of using gen AI and LLM.
  6. The objective, unresolved problem, and key contributions of this work should be written explicitly in separated paragraphs.

Reviewer 3 Report

Comments and Suggestions for Authors

The following comments should be incorporated.

  1. The paper would benefit from a clearer articulation of its unique contribution to the field.
  2. A more detailed exploration of the biological analogues and their computational counterparts would strengthen the conceptual framework.
  3. A deeper synthesis of swarm intelligence principles and biological communication models is necessary.

Reviewer 4 Report

Comments and Suggestions for Authors

The paper presents a novel bio-generative visual communication system for UAV swarms in RF-denied environments, leveraging biologically inspired strategies (e.g., bee waggle dances, deer tail-flagging) and generative AI (LLM with RAG). The work is highly innovative, particularly in its integration of biological communication principles with AI-driven decentralized swarm coordination. The logical flow is strong, with clear sections on motivation, methods, and validation. However, the results, while promising, are limited to simulated test cases, and the discussion could better address scalability and real-world deployment challenges.

  1. The five test cases demonstrate proof-of-concept success in signal interpretation and propagation, validated via covariance analysis. However, conclusions lack depth on real-world constraints (e.g., turbulence, LED visibility). The future work section is comprehensive but could prioritize immediate next steps (e.g., hardware integration).
  2. The results rely on simulated test cases; include pilot experiments with physical UAVs to validate robustness under environmental noise (e.g., wind, partial LED visibility). Example: "Future work should test the system’s resilience to turbulence and sensor occlusions in outdoor deployments."
  3. The near-zero covariance for LED-based signals (Test Cases 3–4) suggests poor interpretability; revise the methodology to enhance LED pattern variability or justify this limitation. Example: "Address why LED signals underperformed and propose dynamic pattern designs (e.g., Morse-like pulses) to improve fidelity."
  4. The paper assumes homogeneous UAVs; analyze how role specialization (heterogeneity) impacts swarm coordination at larger scales. 
  5. Contrast the bio-generative approach with non-AI alternatives (e.g., rule-based optical signaling) to highlight advantages/disadvantages.
  6. The use of GPT-4o may be impractical for edge devices; discuss lightweight LLM alternatives or on-device fine-tuning. 
  7. Address potential misuse (e.g., adversarial spoofing of visual signals) and mitigation strategies. Example: "Discuss how the system authenticates signals to prevent spoofing in contested environments."

Reviewer 5 Report

Comments and Suggestions for Authors

The paper presents a proof of concept that identifies, evaluates, and translates biological communication strategies into a generative visual language for UAV swarms operating in RF-denied environments.

The paper studies an important problem and the ideas are clearly presented.

One main weakness of the paper is the depiction of results in Figures 4-6 and its corresponding explanation.  The results should be better discussed for the readers to follow. The figures in itself are not intuitive and is hard to understand what they represent without reading the detailed explanation. 

Round 2

Reviewer 1 Report

Comments and Suggestions for Authors

The manuscript has been successfully revised. It can be now accepted as is.

Comments on the Quality of English Language

A minor language polishing is necessary.

Author Response

We appreciate the time taken to review the updates made to the manuscript and sincerely thank the reviewer for their positive assessment and recommendation for acceptance. For the final revisions made to the manuscript, we polished the language throughout each section to improve clarity, flow, grammar, and consistency. In addition, we made minor polishing edits to the results section to ensure that the findings are presented with sufficient clarity and phrased effectively for readers to interpret.

Reviewer 2 Report

Comments and Suggestions for Authors

In general, authors have made significant improvement to address the reviewer's suggestions.  The mathematical formulation is now also sufficient. Meanwhile, it would be better if the conclusion is improved once again because it is still too long. Meanwhile, it would also be better if there is any quantitative result in abstract.

Reviewer 3 Report

Comments and Suggestions for Authors

The article can be accepted.

Author Response

We appreciate the time taken to review the updates made to the manuscript and sincerely thank the reviewer for their positive assessment and recommendation for acceptance.

Reviewer 4 Report

Comments and Suggestions for Authors

My questions have been answered well. Thanks to authors.

Author Response

We sincerely thank the reviewer for their constructive feedback during the earlier round and are pleased that the revisions have addressed their questions satisfactorily. For our final revisions to the manuscript, we made minor polishing edits to slightly refine the introduction and to ensure that the background, objective, and research problem are sufficiently clear and well phrased.